# A Study on Secret Key Rate in Wideband Rice Channel

**Simone Del Prete ***[iD], **Franco Fuschini** [iD] **and Marina Barbiroli** [iD]

Department of Electrical, Electronic and Information Engineering "G. Marconi", University of Bologna, 40126 Bologna, Italy
* Correspondence: simone.delprete4@unibo.it

**Abstract:** Standard cryptography is expected to poorly fit IoT applications and services, as IoT devices can hardly cope with the computational complexity often required to run encryption algorithms. In this framework, physical layer security is often claimed as an effective solution to enforce secrecy in IoT systems. It relies on wireless channel characteristics to provide a mechanism for secure communications, with or even without cryptography. Among the different possibilities, an interesting solution aims at exploiting the random-like nature of the wireless channel to let the legitimate users agree on a secret key, simultaneously limiting the eavesdropping threat thanks to the spatial decorrelation properties of the wireless channel. The actual reliability of the channel-based key generation process depends on several parameters, as the actual correlation between the channel samples gathered by the users and the noise always affecting the wireless communications. The sensitivity of the key generation process can be expressed by the secrecy key rate, which represents the maximum number of secret bits that can be achieved from each channel observation. In this work, the secrecy key rate value is computed by means of simulations carried out under different working conditions in order to investigate the impact of major channel parameters on the SKR values. In contrast to previous works, the secrecy key rate is computed under a line-of-sight wireless channel and considering different correlation levels between the legitimate users and the eavesdropper.

**Keywords:** physical layer security; Rice channels; wireless communications; 6G security

## 1. Introduction

Modern cryptography is usually based on mathematical algorithms and can be divided into symmetric and asymmetric encryption systems. Symmetric encryption employs the same key to both encrypt and decrypt messages, while asymmetric cryptography relies on two keys: a public key to turn a plaintext into a ciphertext and a private key to retrieve the plain message. In this framework, the advent of quantum computers might be a threat for modern cryptography systems, that are usually termed to be only computational secure [1]. As an example, RSA, the most popular system for asymmetric cryptography, can easily be broken by Shor's algorithm if run by a quantum computer [2]. Instead, the actual symmetric encryption standard AES, in its version AES-256, is proven to be quantum resistant [3,4]. Moreover, starting from 5G, it is possible to observe a pervasive spread of low-power devices such as IoT devices, which are usually battery powered and have limited computational capacity: the modern RSA system is too lavish to be used on such devices. Therefore, not only is there the need of a quantum resistant set of security techniques, but also methods that can be supported by IoT devices. In this framework, in August 2018, the NIST published a call for an algorithm for lightweight cryptography (https://csrc.nist.gov/projects/lightweight-cryptography, accessed on 12 August 2022), showing the interest from the standardization bodies into the research for new lightweight cryptographic methods.

Physical layer security (PLS) is an umbrella of techniques which is hopefully able to achieve perfect secrecy by exploiting the unpredictable fading characteristics of the wireless channel [1,5]. In addition, PLS has recently been proposed as a key enabler for

the security of future 6G communications systems [6,7]. Among the different techniques fostered under the aegis of PLS, physical layer key generation (PLKG) seems to be a mature and promising solution to protect the confidentiality of communications [8,9], in particular when low-power devices are employed in the system (e.g., IoT devices [10]). PLKG allows two users (here referred to as Alice and Bob) to generate a symmetric encryption key, simply by a mutual observation of the wireless channel, which is desirable to be symmetric and random. In the end, there is a nice interest to employ PLS techniques in the future security paradigm.

Several different metrics have been considered to assess the performance of the PLKG protocol, including key randomness, key disagreement and key generation rate. In particular, the secrecy key rate (SKR) may be of special interest, as it represents the maximum number of bits that can be achieved from each channel observation without the possibility of an eavesdropper (referred to as Eve) catching them [11]. Previous studies on the PLKG have mainly focused on the feasibility of the key generation process under the following conditions:

- The wireless channel is usually considered to be affected by Rayleigh fading, which is only suitable for uniform scattering and non-line-of-sight (LoS) communications.
- The possible presence of an eavesdropper is often neglected, although it represents a real limitation to the number of bits that can be reliably extracted from the channel.
- Alice and Bob are assumed to perceive perfectly symmetric channels, whereas this might not be true under real working conditions, as long as they cannot simultaneously sense the channel for whatever reason. In addition, the channel observations collected by Alice and Bob are affected by the noise and/or hardware imperfections.

These assumptions might not be true in a real scenario: higher frequency (e.g., mm-Wave, Tera-hertz) will be used in the future, already starting from the 5G standards [12]. Therefore, the wireless propagation will likely occur mostly in LOS condition, in order to cope with the high attenuation of the high frequency-bands. Moreover, due to the inner broadcast nature of the wireless channel, it is always possible to eavesdrop the communication, and in this specific case, try to steal some bits of the key.

In the literature, the SKR is usually reduced to the mutual information between Alice and Bob [13], i.e., neglecting the presence of Eve in the channel, who nonetheless decreases the number of bits that can be securely extracted. However, in [14], the authors considered the presence of the eavesdropper, but assumed Gaussian channel samples which might not be true in reality. In addition, it is often assumed that the generation occurs in a non-LoS scenario, i.e., under Rayleigh-like fading conditions [15], which is the ideal case for the PLKG thanks to the high entropy of the channel. Few works in the literature have evaluated the PLKG under LoS conditions, e.g., Ref. [16] computed an upper bound on the key generation capability of the two users communicating under LoS conditions. However, they considered the case in which the eavesdropper is capable of estimating the LoS component, and they assumed perfect channel reciprocity.

The goal of this work is to assess the performance of the PLKG through the computation of the SKR. Monte Carlo simulations have been performed under real-case general conditions with the aim of estimating the SKR in a LoS wireless link. In addition, an eavesdropper (Eve) is assumed to be present and sees the Alice–Bob channel with a low, but not zero, correlation: the correlation matrix of the Alice–Bob, Alice–Eve and Bob–Eve channels is an input parameter of the simulation. Moreover, instead of mutual information, the entire SKR, with its upper and lower bound, is computed. Additionally, the channels are generated according to a realistic 3GPP channel model (as it will be described in Section 3.2). Furthermore, the reciprocity is not assumed to be perfect and the impact of non-ideal reciprocity is taken into account by generating highly correlated channels between the legitimate users, but not equals. The simulations are repeated for different channel conditions: different Rice factor, signal-to-noise ratio, and delay spread (DS).

The rest of the paper is organized as follows: the PLKG protocol is shortly introduced in Section 2. Section 3 explains the assessment simulation procedure, whereas Section 4

reports a validation of the simulation procedure under a reference Gaussian case. The results of the assessment are reported in Section 5 and finally some conclusions are drawn in Section 6.

## 2. Physical-Layer Key Generation Protocol

The aim of the PLKG protocol is to let Alice and Bob autonomously generate a symmetric encryption key, without the possibility for Eve to steal the key. It fundamentally relies on the following general properties of the propagation channel [1]:

- **Reciprocity**: it is known that the wireless channel is almost symmetric between the transmitter and the receiver: this allows Alice and Bob to obtain similar channel samples, by means of mutual and possibly simultaneous observations of the channel.
- **Randomness**: mostly because of the fast fading, the channel fluctuates in time, frequency or space domain, in a random-like fashion, which ensures extracting random-like keys from the channel.
- **Spatial decorrelation**: provided that Eve is placed at a large distance (with respect to the wavelength), she will observe uncorrelated channel samples from Alice/Bob observations, which can strongly limit the actual eavesdropping threat.

PLKG usually consists of four well-known stages [1]:

1. **Channel probing**: Alice and Bob should almost simultaneously sample the channel in order to extract some channel features. The observation pairs must be retrieved within the coherence time of the fading in order to have similar channel samples. Moreover, different features can be extracted: channel observations are often limited to the received signal strength indicator (RSSI), but the whole channel state information (CSI) can also be targeted. Despite being more difficult to obtain, the CSI provides a wider set of features, which leads to an higher number of bits that can be possibly extracted.

2. **Quantization**: since the channel features are analog values, they must be quantized according to some quantization scheme to obtain, at the end of the process, digital encryption keys. A different quantization scheme can be used: uniform or non-uniform, single- or multi-level, differential-based or mean value-based [1,17].

3. **Information Reconciliation**: due to possible imperfection in the hardware, due to noise and non-ideal channel reciprocity, the quantized values might be slightly different between Alice and Bob. These discrepancies can be settled by means of standard error correction techniques through the exchange of public messages between Alice and Bob [18].

4. **Privacy amplification**: at the end of the previous stages, Eve might have acquired some bits of the key agreed between Alice and Bob, depending on the degree of randomness and spatial decorrelation inside the propagation channel. Privacy amplification is therefore enforced, where a new key is distilled by applying a randomly selected Hash function to the key achieved after the reconciliation procedure. Thanks to the properties of the universal hash function, the final key is likely to be fully unknown to Eve [19].

An important metric for the PLKG is the SKR, which was introduced by Maurer in [11]. Suppose that Alice, Bob and Eve, respectively, acquire the channel observations $X^A = [x^A(1), x^A(2), \cdots, x^A(n)]$, $X^B = [x^B(1), x^B(2), \cdots, x^B(n)]$, $X^E = [x^E(1), x^E(2), \cdots, x^E(n)]$, then the SKR has an upper and lower bound expressed by [11]:

$$\mathbb{R}(X^A, X^B \parallel X^E) \geq \max[\mathbb{I}(X^A; X^B) - \mathbb{I}(X^A; X^E), \mathbb{I}(X^A; X^B) - \mathbb{I}(X^B; X^E)], \tag{1}$$

$$\mathbb{R}(X^A, X^B \parallel X^E) \leq \min[\mathbb{I}(X^A; X^B), \mathbb{I}(X^A; X^B \mid X^E)], \tag{2}$$

which is an indication of the maximum number of bit per channel observation that can be extracted without the possibility of Eve guessing the bit [1]. The presence of Eve is taken into consideration in this work, as the information leakage to a possible eavesdropper can



actually further limit the SKR. Furthermore, the channel is not assumed to be perfectly symmetric: the channel observations of Alice and Bob are still highly correlated, but not exactly the same. In order to compute the SKR under complete working conditions, a Monte Carlo simulation was therefore carried out, as explained in Section 3.

## 3. Materials and Methods

The main goal of the work is to assess the value of the SKR under different channel conditions in a system where the encryption keys are generated according to the previously explained PLKG protocol. Figure 1 outlines the presence of the users in the channel, with a particular emphasis on the mutual correlation, whereas a summary of the simulation parameters is reported in Table 1. The target observation is the frequency response of the channel, processed through the filterbank method [20]. Therefore, the vectors of channel observations $X^A$, $X^B$, $X^E$ consist of the output of the $N_f$ filters applied to the power spectral density (PSD). Moreover, the filters are supposed to be ideal pass band filters and the PSD is obtained through the square FFT of the channel impulse response (CIR), which is generated according to a wideband tapped delay model [21], where it is possible to tune the delay spread (DS) and the Rice factor K. Furthermore, the PSD observed by Alice Bob, and Eve is generated according to some mutual correlation target. This is accomplished through the Cholesky decomposition, even though it is only theoretically supported in the case of Gaussian samples. The channels are generated in order to achieve a bandwidth of 160 MHz.

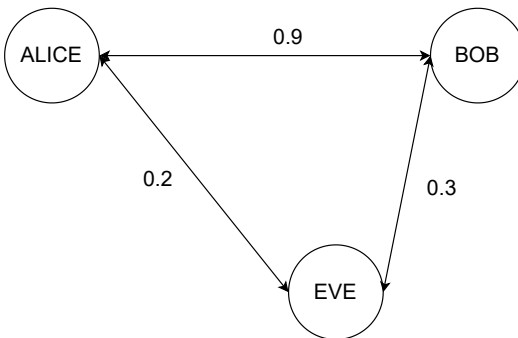

**Figure 1.** General scheme of Alice, Bob and Eve on the channel: each pair of users sees the channel realizations with a different non-zero correlation.

**Table 1.** Main simulation parameters.

| Parameter | Value |
|---|---|
| Bandwidth (MHz) | 160 |
| Sampling time | $2 \times 10^{-9}$ s |
| SNR reference value (dB) | 10 |
| SNR (dB) | From 0 to 30 with step of 2 |
| Channel realizations | 50,000 |
| Nfft | 2048 |
| Number of filters | 1 or 4 |
| Delay spread reference value (ns) | 30 |
| Delay spread (ns) | [10, 30, 100, 300, 600, 1000, 2000, 5000] |
| K reference value (dB) | 10 |
| K array (dB) | from 0 to 30 with step of 2 |
| Alice–Bob correlation | 0.99, 0.9, 0.7 |
| Alice–Eve/Bob–Eve correlation | 0.1, 0.2, 0.7 |

The SKR is computed through a Monte Carlo simulation: $5 \times 10^5$ channel realizations are generated for the same input values (DS and Rice factor K) and the SKR is computed case by case according to (1) and (2). The different simulation steps are described in the following sections, and a scheme of the procedure is sketched in Figure 2.

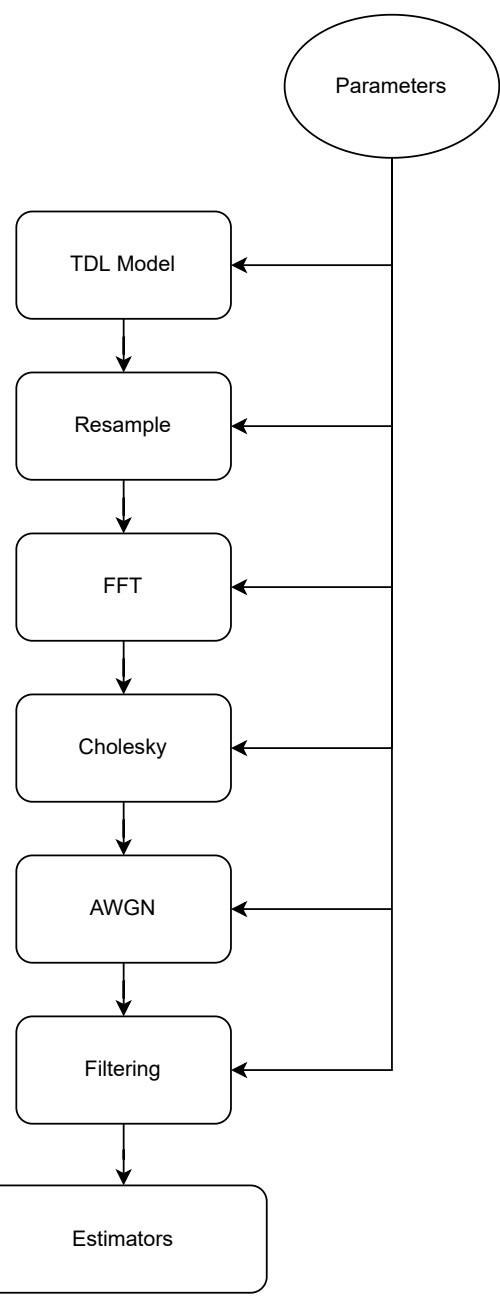

**Figure 2.** Diagram of the simulation.

*3.1. Parameters*

The first block in the simulation flow chart outlined in Figure 2 refers to a parameter file listing the parameters required by each simulation snapshot. The main parameters are reported in Table 1.

*3.2. Tapped Delay Line Model*

The wireless channel is generated according to the Tapped Delay Line "TDL-D" model described in [21]. It is a statistical channel model and consists of a set of paths with a normalized delay and power, which can be tuned to account for different propagation conditions. In particular, the channel model accounts for multipath Rice fading, i.e., the Rice factor and the DS are the tuning parameters of the model.

To generate the channel realizations, the following procedure was applied, as also described in [21]:

1.  Modify the power and the delays of the TDL according to the procedure described on page 83 of [21], in order to have a given Rice factor and a DS;
2.  As for the first line of the TDL, the component is generated as a Rice random variable with a K-factor equal to the desired one: this represents the LOS component of the channel.
3.  For each multipath line, generate a complex Gaussian random variable with a zero-mean and a variance equal to the mean power of each line. As such, it is possible to generate Rayleigh-fading lines with a mean power specified by the average received power of each line of the TDL.

### 3.3. Resample

The TDL model is then resampled in order to obtain a CIR with a continuous time axis. To this aim, a sample time is selected as the inverse of the channel bandwidth written in Table 1. Each delay of the TDL is transformed into the corresponding time sample, and the complex amplitudes of the taps falling within the same sample are coherently summed up.

### 3.4. FFT

To obtain the channel frequency response, a simple FFT is performed on the CIR, which is also zero padded to reach "Nfft" samples (see Table 1). For the purpose of this work, the square amplitude of the channel transfer function (CTF), often referred to as Power Spectral Density (PSD) is considered. Therefore, the filtering applies to the PSD.

### 3.5. Cholesky Decomposition

Cholesky decomposition is a matrix decomposition procedure often employed to generate correlated Gaussian samples. Let $\underline{X} = (x_1, x_2, \dots x_n)$ be a $n$-dimensional standard Gaussian random vector ($x_i \sim \mathcal{N}(0,1)$) made of uncorrelated samples: its covariance matrix will be the identity matrix. A set of correlated Gaussian random variables can be obtained through the Cholesky decomposition, which decomposes an Hermitian matrix ($\overline{\overline{C}}$) into the product of a triangular lower ($\overline{\overline{L}}$) and a triangular upper matrix ($\overline{\overline{L}}^T$).

$$\overline{\overline{C}} = \overline{\overline{L}} \times \overline{\overline{L}}^T.$$
(3)

The vector $\overline{Y} = \overline{\overline{L}} \times \overline{X}$ will then be a Gaussian random vector with a covariance matrix equal to $\overline{\overline{C}}$. The proof of this is simple and follows from the computation of the covariance matrix of $\overline{Y}$:

$$E[\overline{Y} \times \overline{Y}^T] = E[\overline{\overline{L}} \times \overline{X} \times (\overline{\overline{L}} \times \overline{X})^T] = E[\overline{\overline{L}} \times \overline{X} \times \overline{X}^T \times \overline{\overline{L}}^T] =$$
$$= \overline{\overline{L}} \times E[\overline{X}\overline{X}^T] \times \overline{\overline{L}}^T = \overline{\overline{L}} \times \overline{\overline{I}}_n \times \overline{\overline{L}}^T = \overline{\overline{L}} \times \overline{\overline{L}}^T = \overline{\overline{C}}.$$
(4)

This method is known to be theoretically grounded for Gaussian variables and according to [22], it is still reliable in case the variables are Gamma distributed. The Rice distribution is approximated by the Nakagami-m distribution and Gamma variables can be obtained as the square of Nakagami-m variables. By means of the Fitter (https://pypi.org/project/fitter/, accessed on 12 August 2022) class, the PSD samples were fitted in order to empirically determine the distribution of the samples. By looking at Figure 3 and Table 2, where the Sumsquare error and the parameters (following the scipy.stats (https://docs.scipy.org/doc/scipy/tutorial/stats.html, accessed on 12 August 2022) notation) is reported for different distributions, the PSD samples distribution seems to fairly comply with a gamma distribution. Therefore, it is reasonable to suppose that the PSD samples are gamma-distributed and the method of the Cholesky decomposition is still reliable in this case. For instance, by setting the target correlation between Alice and Bob to 0.99 and the correlation between Alice/Bob and Eve to 0.1, the actual correlation levels were

then computed from on the channel samples achieved after the Cholesky decomposition, and turned out equal to 0.99 and 0.09.

**Table 2.** Sumsquare error and parameters of different distributions.

| Distribution | Sumsquare Error | Parameters |
|---|---|---|
| gamma | 0.008232 | a = 42.463, loc = −1.146, scale = 0.047 |
| lognorm | 0.008412 | s = 0.102, loc = −2.142, scale = 3.008 |
| chi2 | 0.008457 | df = 34.237, loc = −0.454, 0.038 |
| norm | 0.190884 | loc = 0.881, scale = 0.311 |
| rayleigh | 1.621860 | loc = 0.388, scale = 0.412 |

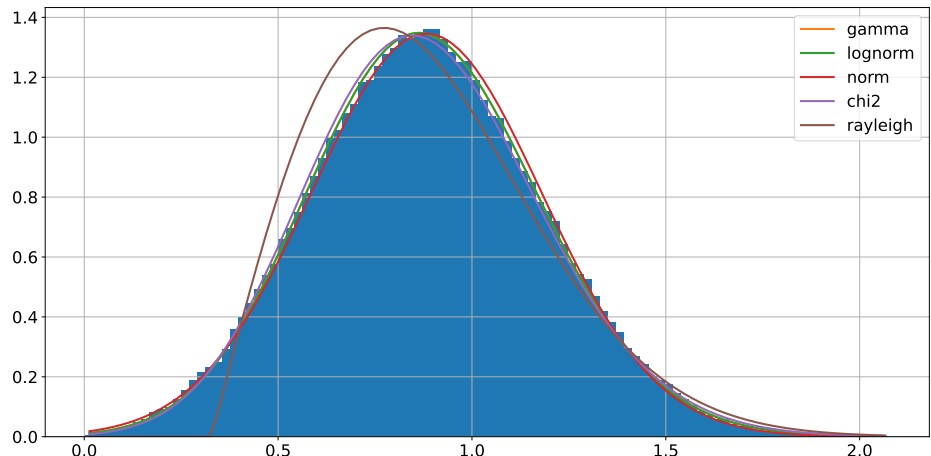

**Figure 3.** Fitting of different probability density functions to the histogram of the PSD samples.

If the matrix $\overline{\overline{C}} = \overline{\overline{L}} \times \overline{\overline{L}}^T$ is the desired correlation matrix, $A_i'$, $B_i'$, $E_i'$ are respectively Alice's, Bob's and Eve's independent $i$-th realization of the PSD, the correlated channels $(A_i, B_i, E_i)$ are obtained through s simple matrix multiplication:

$$\begin{bmatrix} a_{i;0} & \cdots & a_{i;M} \\ b_{i;0} & \cdots & b_{i;M} \\ e_{i;0} & \cdots & e_{i;M} \end{bmatrix} = \overline{\overline{L}} \times \begin{bmatrix} a_{i;0}' & \cdots & a_{i;M}' \\ b_{i;0}' & \cdots & b_{i;M}' \\ e_{i;0}' & \cdots & e_{i;M}' \end{bmatrix} \tag{5}$$

As an example, Figure 4 depicts an example of channel realization, showing that, for high Alice–Bob correlation, the channels in frequency are quite similar, and instead Eve observes an uncorrelated channel.

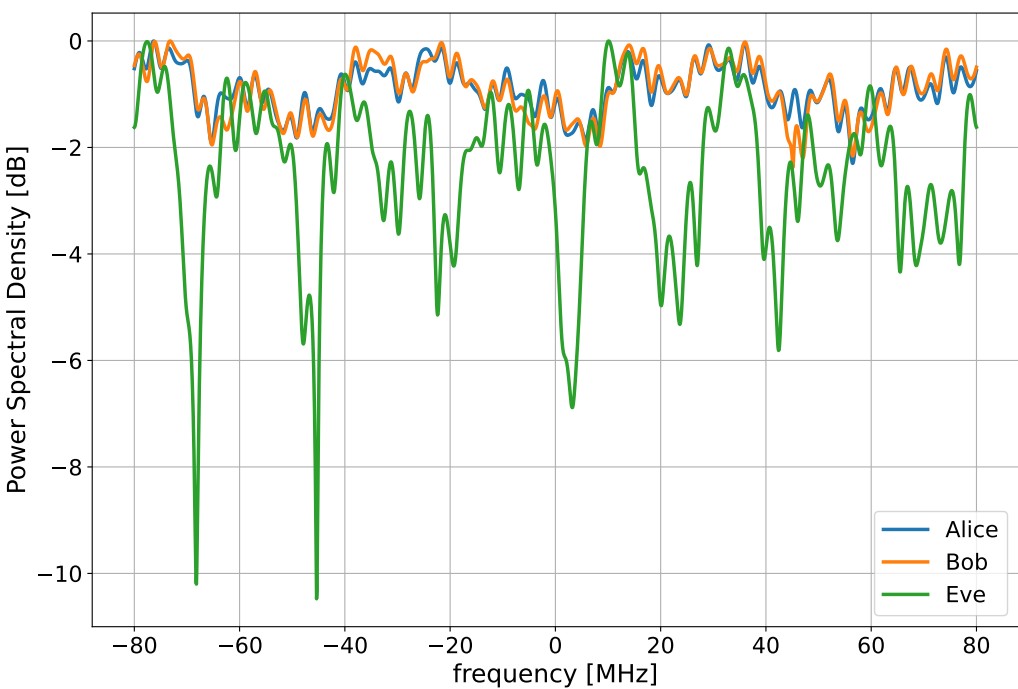

**Figure 4.** A channel realization obtained with K = 10 dB, delay spread = 30 ns, Alice–Bob correlation of 0.99, Alice–Eve and Bob–Eve correlation of 0.2.

### 3.6. AWGN

After the correlation of the channel, white noise is added to the PSD according to the signal-to-noise ratio reported in Table 1.

### 3.7. Filtering

The SKR is computed on the PSD after the filterbank [20] method is applied. For the purpose of this project, the filters are assumed to be ideal pass-band filters and there are either 1 or 4 filters. Each filter acts as a mean operator on the sub band of the PSD (or the entire PSD in case 1 filter is employed), hence the output of a filter is a single number. In practice, if $P(f)$ is the PSD, $f_i$ is the central frequency of the $i$-th filter and $\Delta f$ its pass band, then the output of the filter is computed as follows:

$$X_i = \frac{1}{\Delta f} \int_{f_i - \Delta f/2}^{f_i + \Delta f/2} P(f) \, df, \quad i = 1, 2, \ldots, N_f. \tag{6}$$

The filtering is also useful to reduce the dimensionality of the CTF, which is a benefit for the mutual information estimators, as will be clear in the next paragraph. In case 1 filter is employed, the entire 160 MHz is used; instead, when 4 filters are used, each filter has a non overlapping bandwidth of 40 MHz.

### 3.8. Estimators

Mutual information estimators have been employed to obtain the mutual information required for the computation of the SKR. In particular, the Non-Parametric Entropy Estimator Toolbox (https://github.com/gregversteeg/NPEET, accessed on 15 May 2022) and a python open source estimator of the mutual information based on the channel samples vectors were exploited. Moreover, this allows to estimate the mutual information for a multidimensional sample. However, these kind of estimators requires an exponential number of samples as the dimensionality increases due to the problem known as the curse of dimensionality [23]: therefore, the number of dimensions (number of filters of the filterbank) must be kept low. For the purpose of this work, it was seen that, by using 500,000 channel realizations, the estimators already converge.

## 4. Gaussian Case and Validation

A preliminary assessment was carried out in the Gaussian case, as the mutual information between Gaussian vectors can be expressed through analytical, closed-form formulas. The goal of this section is to evaluate the effectiveness of the simulator in a case where the mutual information can be expressed by an analytical closed formula. In particular, we derived the expression of the mutual information between correlated Gaussian variables and verified the correctness of the method implemented, particularly of the estimators.

Consider two Gaussian signals affected by AWGN:

$$A = s_a + n_a, \tag{7}$$

$$B = s_b + n_b, \tag{8}$$

where $s_a, s_b \sim \mathcal{N}(0,1)$, $n_a \sim \mathcal{N}(0,\sigma_a)$ and $n_b \sim \mathcal{N}(0,\sigma_b)$ and $\mathrm{corr}(s_a, s_b) = \eta$. Since $A$ and $B$ are the sum of a zero mean Gaussian random variable, they will both be Gaussian with a variance, respectively, $\sigma_A$ and $\sigma_B$. The mutual information between $A$ and $B$ can be therefore expressed as:

$$\mathbb{I}(A;B) = h(A) + h(B) + h(A,B) = \frac{1}{2}\log_2\left(\frac{\sigma_A^2 \sigma_B^2}{\sigma_A^2 \sigma_B^2 - \eta^2}\right), \tag{9}$$

See Appendix A for the demonstration.

*Estimation Procedure*

In order to test the estimators, the following procedure is employed. First, independent Gaussian signals are generated, then a correlation is applied according to what has been explained in Section 3.5. After the generation, AWGN is added to the signals:

$$\overline{X_1} \sim \mathcal{N}(0,1), \tag{10}$$

$$\overline{X_2} \sim \mathcal{N}(0,1), \tag{11}$$

$$\overline{n_a} \sim \mathcal{N}(0,\sigma_a), \tag{12}$$

$$\overline{n_b} \sim \mathcal{N}(0,\sigma_b), \tag{13}$$

$$\overline{s_a} = \overline{X_1}, \tag{14}$$

$$\overline{s_b} = \eta\overline{X_1} + \sqrt{1-\eta^2}\,\overline{X_2}, \tag{15}$$

$$\overline{A} = \overline{s_a} + \overline{n_a}, \tag{16}$$

$$\overline{B} = \overline{s_b} + \overline{n_b}. \tag{17}$$

Equation (15) comes from (3) and (5) when two random vectors are considered. The evaluation is repeated for different values of the correlation $\eta$: after the generation, the random vectors are given to the estimators to obtain mutual information. Furthermore, $\overline{X_1}$ and $\overline{X_2}$ contain 500,000 samples.

Figure 5 shows the results of the comparison. In particular, the mutual information significantly drops when the correlation is different from 1. Moreover, the estimated curves correspond to the theoretical case, confirming the correct behavior of the estimators. Since the SKR is a combination of mutual information, the same agreement between the theory and the simulation is expected regarding the SKR. This also proves the correctness of the simulation procedure employed.

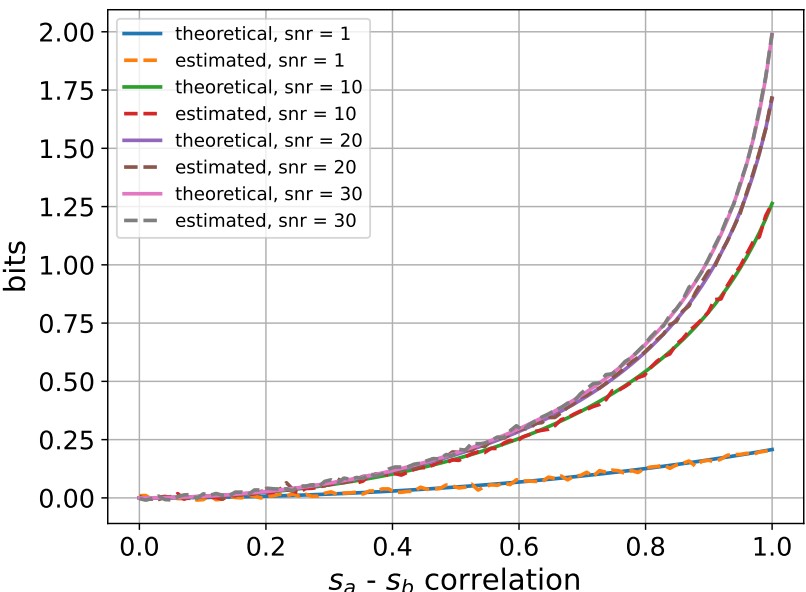

**Figure 5.** Comparison between the theoretical and estimated mutual information in the correlated Gaussian case, with different SNR conditions.

## 5. Results and Discussion

Simulations aimed at evaluating the SKR under different channel conditions, i.e., for different values of the Rice factor, of the DS and of the SNR. For the sake of simplicity, the legitimate and the eavesdropped channels are assumed to share the same Rice factor and DS, and Eve is supposed to have the same correlation towards Alice and Bob indifferently.

### 5.1. SKR and the K Factor

Simulations were run for different values of the Rice factor and correlation between the wireless channels, but always with the same SNR of 10 dB and with a DS of 30 ns. In addition, the estimation was performed both for the one-filter (narrow-band case, Figure 6) and for four-filters (wide-band case, Figure 7) cases. When Alice and Bob share highly correlated channel observations (0.99 in Figures 5 and 6), the SKR lower and upper bound basically coincide: this is not surprising as the lower and upper bound set on the SKR by (1) and (2) come to coincide as soon as Alice and Bob share highly correlated channel observations. Further details can be found in Appendix B. Instead, when the correlation is reduced, the two curves become distinguishable. Moreover, it is possible to highlight a decreasing trend of the SKR with the Rice factor: for a larger *K*, the channels are more stable and the multipath effects are reduced, thus the channel fluctuations are weaker, the overall randomness inside the channel is lower and hence the SKR is reduced. The reasons for this decreasing evolution of the SKR can be found by looking at Figure 8, which reports some PSD for the different values of the Rice Factor. As K increases, the channels become flatter, resulting in a weaker entropy and hence, in a lower SKR.

Reducing the Alice–Bob correlation also impairs the SKR, as it means that the disagreements in the bit sequences harvested from the channel become more probable because of the lower reciprocity level. A further reduction in the SKR is triggered when Eve improves her correlation with respect to Alice/Bob, as she can then better infer some information about the key, thus reducing its overall secrecy. Since the SKR represents the total number of bits that can be extracted after the filterbank method, it is normal to observe higher values when four filters are employed (Figure 7).

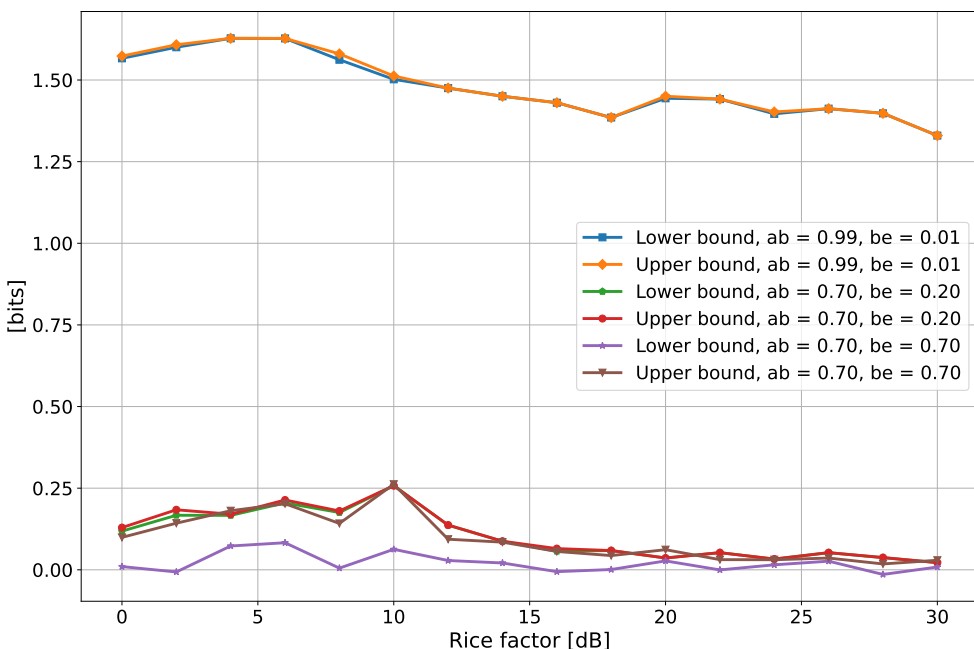

**Figure 6.** Secrecy key rate as a function of the Rice factor K, for different values of the correlation and with 1 filter. In the legend, "ab" and "be" stand for Alice–Bob correlation and Bob–Eve correlation.

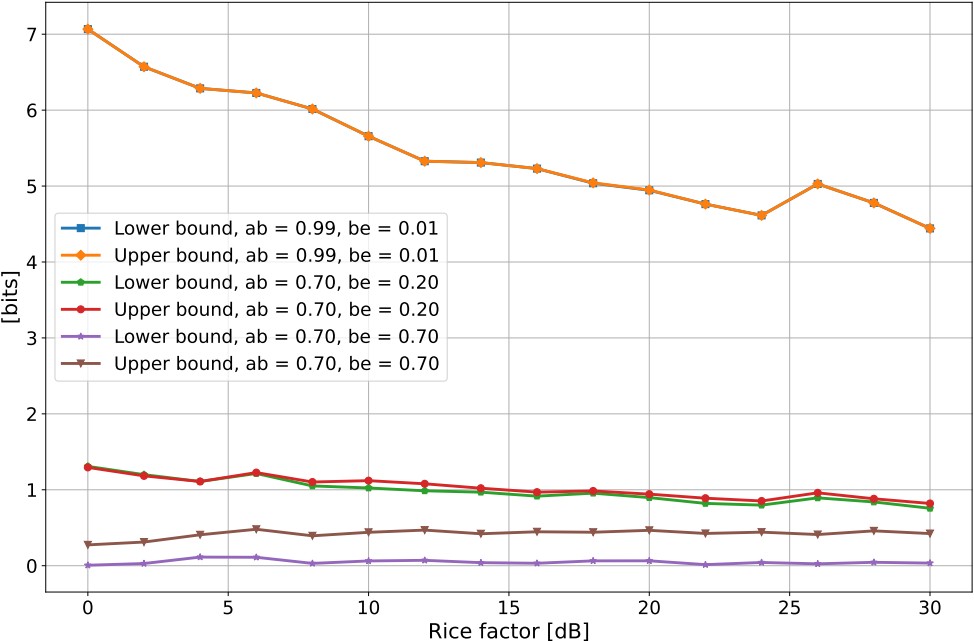

**Figure 7.** Secrecy key rate as a function of the Rice factor K, for different values of the correlation and with 4 filters. In the legend, "ab" and "be" stand for the Alice–Bob correlation and Bob–Eve correlation.

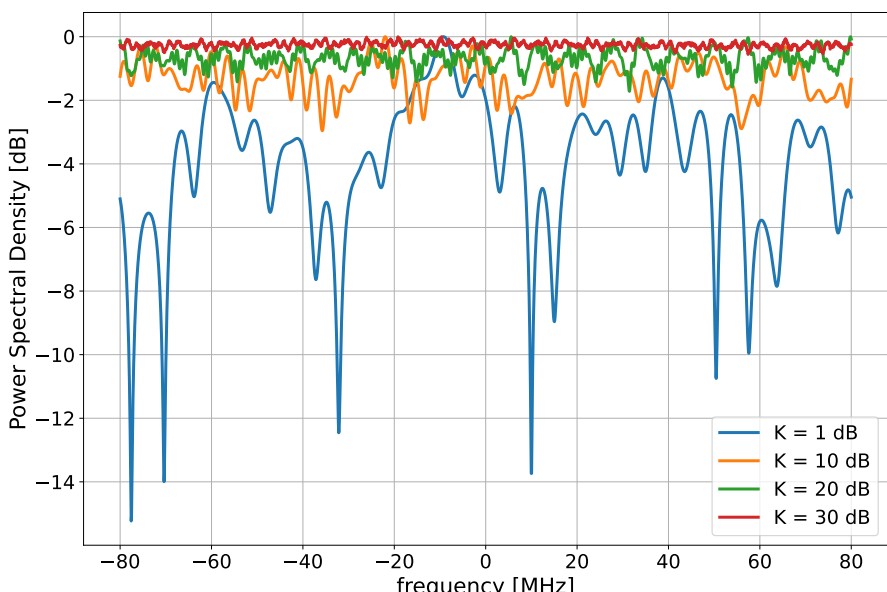

**Figure 8.** Power spectral densities for a different value of the Rice factor K.

### 5.2. SKR and SNR

The simulations were then performed with respect to the SNR experienced by Alice and Bob, whereas the SNR of Eve is always kept to 10 dB, the DS is 30 ns and the Rice factor was set to 10 dB. Once again, the simulations were repeated for different values of the correlation.

Figure 9 depicts the SKR as a function of the SNR with one filter, while Figure 10 shows the situation with four filters. In line with the Gaussian case described in Section 4, the SKR increases with the SNR, as a louder noise between Alice and Bob evidently affects the channel reciprocity, thus increasing the probability of disagreement between the key they finally receive from the channel observations. The sensitivity to the channels' correlation highlighted in Figures 9 and 10 is of course the same as that already discussed with reference to Figures 6 and 7.

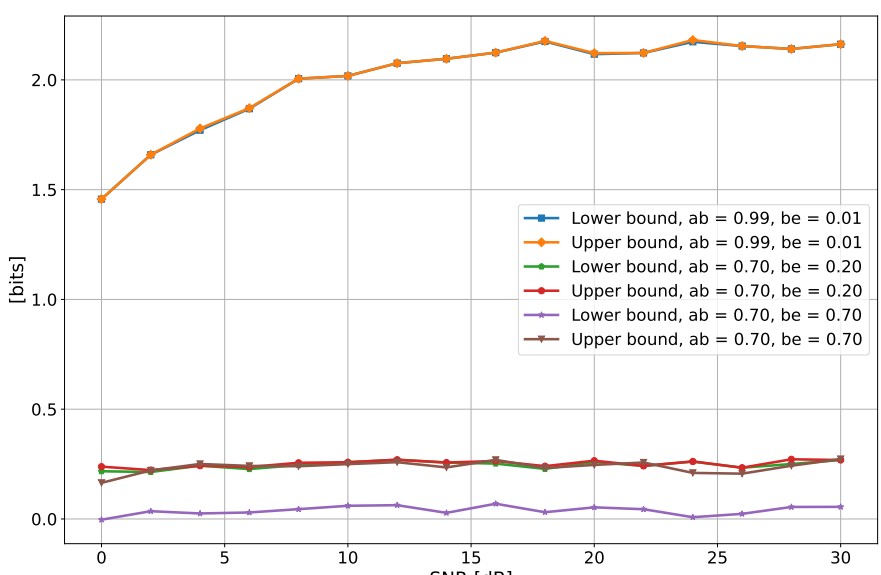

**Figure 9.** Secrecy key rate as a function of the SNR of Alice and Bob, for different values of the correlation and with 1 filter. In the legend, "ab" and "be" stand for Alice–Bob correlation and Bob–Eve correlation.

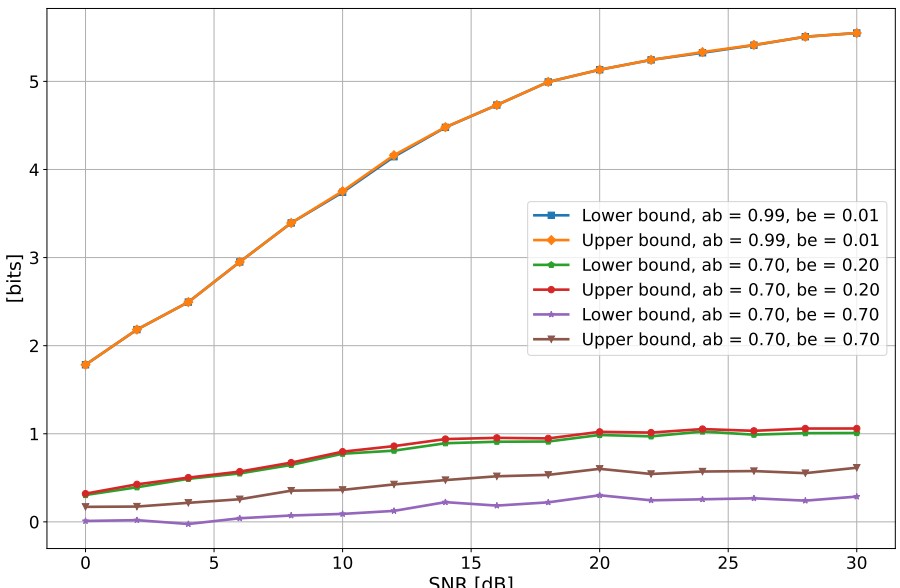

**Figure 10.** Secrecy key rate as a function of the SNR of Alice and Bob, for different values of the correlation and with 4 filters. In the legend, "ab" and "be" stand for Alice–Bob correlation and Bob–Eve correlation.

### 5.3. SKR and Delay Spread

As a last case, the simulations were performed to fix both the SNR and the Rice factor at 10 dB, but varying the DS of the channel. As in the previous cases, the simulations are repeated for different values of the correlation values.

As for the case with one filter, depicted in Figure 11, it is possible to notice that the DS does not seem to have a big impact on the SKR. Conversely, the SKR tends to decrease with the increasing DS, when multiple filters are employed (Figure 12). This trend is also in line with what has been reported in [13].

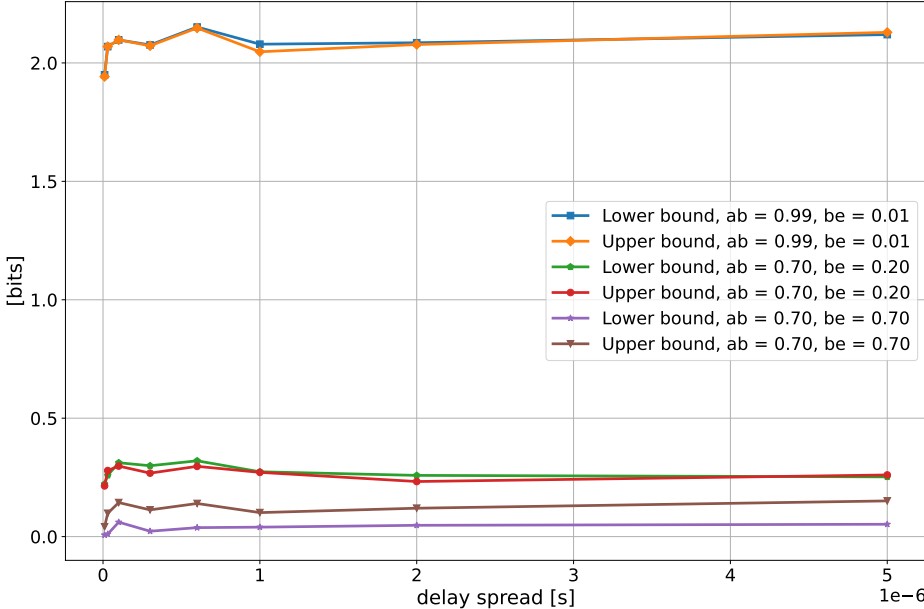

**Figure 11.** Secrecy key rate as a function of the delay spread of the channel of Alice and Bob, for different values of the correlation and with one filter. In the legend, "ab" and "be" stand for the Alice–Bob correlation and Bob–Eve correlation.

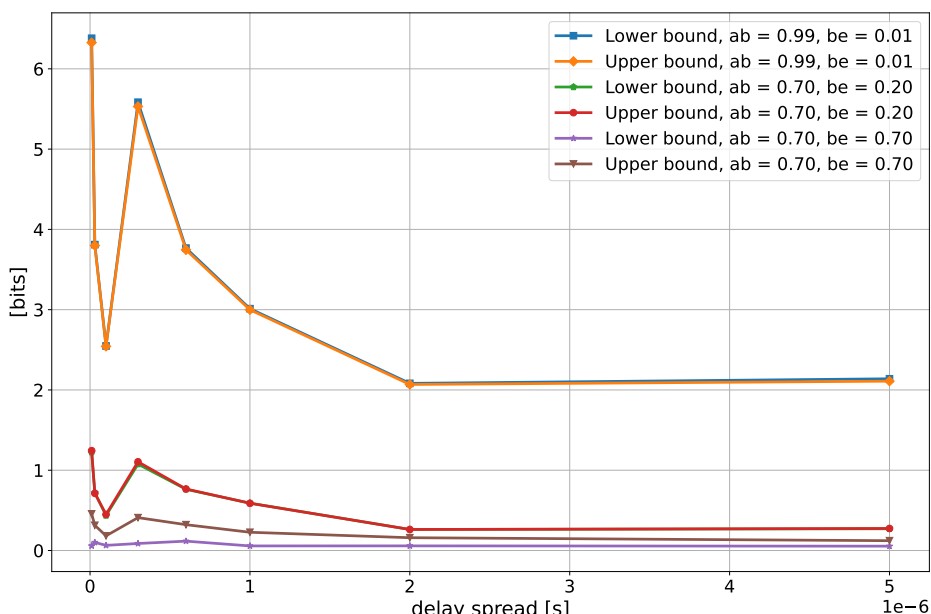

**Figure 12.** Secrecy key rate as a function of the delay spread of the channel of Alice and Bob, for different values of the correlation and with 1 filter. In the legend, "ab" and "be" stand for Alice–Bob correlation and Bob–Eve correlation.

The reason for this behavior can be understood by looking at Figure 13 and bearing in mind that the number of paths in the TDL is fixed: when the DS is low, there is a higher probability that the different paths cannot be resolved singularly; therefore, they might severely interfere and create a deep null in the PSD. In contrast, when the DS is larger, the different paths are spread over a wider delay range, and therefore they less frequently add up coherently inside the PSD, thus corresponding to a more oscillating PSD, but without deep fades.

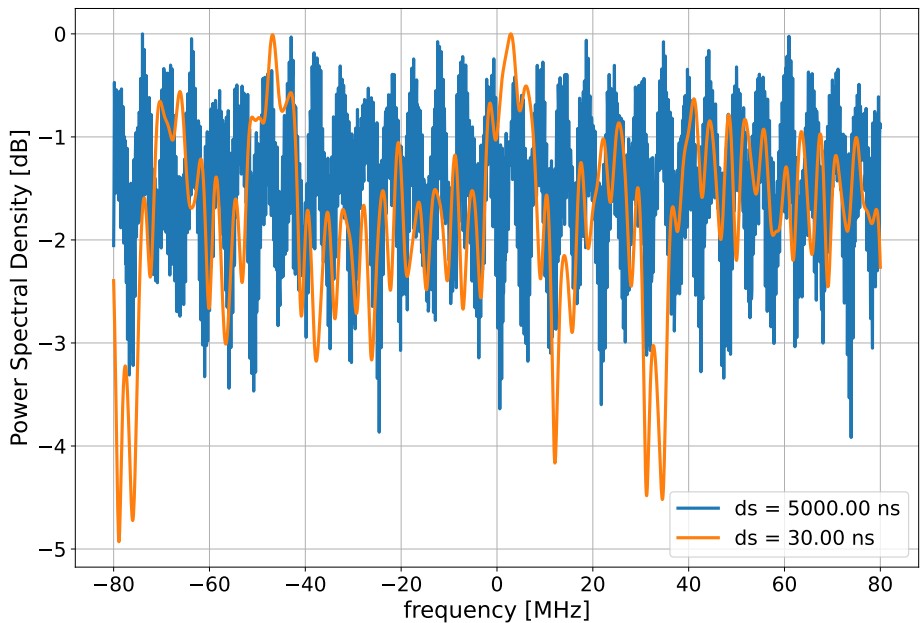

**Figure 13.** A realization of power spectral density with a different delay spread.

In terms of the entropy of the channel, and hence mutual information between Alice and Bob, having deep fades increases the randomness of the channel, translating into a

higher SKR. Moreover, the effect of the deep fades is somehow mitigated in the case of one single filter, since it blunts the effects due to the presence of deep fades by averaging the PSD over the whole signal bandwidth. Instead, when four filters are employed, the deep fades in the case of low DS create more variability on the filter outputs, introducing more entropy.

## 6. Conclusions

In this work, a simulation framework for PLKG in the Rice channel was presented, in order to compute the SKR under different channel conditions. Moreover, the simulator is able to generate correlated wide-band channel in order to take into account the presence of an eavesdropper and the possible imperfections that lead to non-ideal channel reciprocity. The SKR was computed, showing a decreasing trend with respect to the Rice factor of the channel. Moreover, it was shown that a high correlation between the Alice and Bob channel samples is required in order to achieve a reasonable SKR. Finally, given the considered channel model, the DS has a detrimental effect on the SKR, since the higher DS situations lead to a lower SKR.

**Author Contributions:** Conceptualization, S.D.P.; methodology, S.D.P.; software, S.D.P.; validation, S.D.P., F.F. and M.B.; investigation, S.D.P.; writing—original draft preparation, S.D.P. and F.F.; writing—review and editing, M.B. All authors have read and agreed to the published version of the manuscript.

**Funding:** This research received no external funding.

**Institutional Review Board Statement:** Not applicable.

**Informed Consent Statement:** Not applicable.

**Data Availability Statement:** Data and software available under request.

**Conflicts of Interest:** The authors declare no conflict of interest.

## Appendix A

Suppose we have two signals with AWGN:

$$A = s_a + n_a \tag{A1}$$

$$B = s_b + n_b \tag{A2}$$

where $s_a, s_b \sim \mathcal{N}(0,1)$, $n_a \sim \mathcal{N}(0, \sigma_a)$ and $n_b \sim \mathcal{N}(0, \sigma_b)$.

If $SNR$ is the signal-to-noise ratio of the two users (assumed to be the same for simplicity):

$$\sigma_{n_a}^2 = \frac{1}{SNR}, \tag{A3}$$

$$\sigma_{n_b}^2 = \frac{1}{SNR}, \tag{A4}$$

$$\sigma_a^2 = 1 + \frac{1}{SNR}, \tag{A5}$$

$$\sigma_b^2 = 1 + \frac{1}{SNR}. \tag{A6}$$

Now, suppose that $s_a$ and $s_b$ have a covariance $\text{cov}(s_a, s_b) = \eta$: since the variance of both $s_a$ and $s_b$ is equal to 1, then the covariance and the correlation are the same. The correlation between $A$ and $B$ can be computed as:

$$\text{corr}(A,B) \equiv \rho = \frac{E[(A - \mu_A)(B - \mu_B)]}{\sigma_A \sigma_B}, \tag{A7}$$

but $\mu_A = \mu_B = 0$ since they are the sum of the zero mean Gaussian random variables, therefore

$$\text{corr}(A, B) \equiv \rho = \frac{E[(A - \mu_A)(B - \mu_B)]}{\sigma_A \sigma_B} = \frac{E[AB]}{\sigma_A \sigma_B} = \frac{E[(s_a + n_a)(s_b + n_b)]}{\sigma_A \sigma_B} =$$

$$= \frac{E[s_a s_b] + E[s_a n_b] + E[s_b n_a] + E[n_a n_b]}{\sigma_A \sigma_B} = \frac{E[s_a s_b]}{\sigma_A \sigma_B} = \quad \text{(A8)}$$

$$= \frac{\text{cov}(s_a, s_b)}{\sigma_A \sigma_B} = \frac{\eta}{\sigma_A \sigma_B}.$$

The mutual information of the two random variables can be rewritten as

$$\mathbb{I}(A; B) = h(A) + h(B) + h(A, B), \quad \text{(A9)}$$

where $h(A)$, $h(B)$ are the differential entropies of the two signals and $h(A, B)$ is the joint entropy, which in the Gaussian case, can be expressed as:

$$h(A) = \frac{1}{2} \log_2(2\pi e \sigma_A^2), \quad \text{(A10)}$$

$$h(B) = \frac{1}{2} \log_2(2\pi e \sigma_B^2), \quad \text{(A11)}$$

$$h(A, B) = \frac{1}{2} \log_2\left((2\pi e)^2(\sigma_A^2 \sigma_B^2 - \text{cov}^2(A, B))\right) = \quad \text{(A12)}$$

$$= \frac{1}{2} \log_2\left((2\pi e)^2(\sigma_A^2 \sigma_B^2 - \sigma_A^2 \sigma_B^2 \rho^2)\right).$$

Hence, the mutual information can be written as:

$$\mathbb{I}(A; B) = h(A) + h(B) + h(A, B) =$$

$$= \frac{1}{2} \log_2(2\pi e \sigma_A^2) + \frac{1}{2} \log_2(2\pi e \sigma_B^2) - \frac{1}{2} \log_2\left((2\pi e)^2(\sigma_A^2 \sigma_B^2 - \sigma_A^2 \sigma_B^2 \rho^2)\right) =$$

$$= \frac{1}{2} \log_2\left(\frac{(2\pi e)^2 \sigma_A^2 \sigma_B^2}{(2\pi e)^2(\sigma_A^2 \sigma_B^2 - \sigma_A^2 \sigma_B^2 \rho^2)}\right) = \frac{1}{2} \log_2\left(\frac{\sigma_A^2 \sigma_B^2}{\sigma_A^2 \sigma_B^2 - \sigma_A^2 \sigma_B^2 \rho^2}\right) = \quad \text{(A13)}$$

$$= \frac{1}{2} \log_2\left(\frac{1}{1 - \rho^2}\right) = \frac{1}{2} \log_2\left(\frac{\sigma_A^2 \sigma_B^2}{\sigma_A^2 \sigma_B^2 - \eta^2}\right).$$

**Appendix B**

Let's start from the Lower Bound, assuming that Alice and Bob share highly correlated channel samples and that the correlation between Alice and Eve channel samples is the same as that between the Bob and Eve samples. Moreover, all the links share the same channel condition in terms of SNR and Rice factor K. The lower bound of the SKR can be reduced to:

$$\mathbb{R}(X^A, X^B \parallel X^E) \geq \max[\mathbb{I}(X^A; X^B) - \mathbb{I}(X^A; X^E), \mathbb{I}(X^A; X^B) - \mathbb{I}(X^B; X^E)]$$

$$= \mathbb{I}(X^A; X^B) - \mathbb{I}(X^A; X^E) \quad \text{(A14)}$$

$$= h(X^A) - h(X^A | X^B) - h(X^A) + h(X^A | X^E) \simeq h(X^A | X^E).$$

The conditioned entropy $h(X^A | X^B)$ is almost zero since the Alice and Bob channel observations are highly correlated, and therefore, the residual uncertainty on $X^A$ by knowing $X^B$ is almost null.

In the same way as before, the upper bound can be reduced to:

$$\mathbb{R}(X^A, X^B \parallel X^E) \leq \min[\mathbb{I}(X^A; X^B), \mathbb{I}(X^A; X^B \mid X^E)] =$$

$$= \min[h(X^A) - h(X^A | X^B), h(X^A | X^E) - h(X^A | X^B, X^E)] \quad \text{(A15)}$$

$$\simeq \min[h(X^A), h(X^A | X^E)] = h(X^A | X^E).$$

The term $h(X^A|X^B)$ is zero for the reasons explained before, and then the term $h(X^A|X^B, X^E)$ is almost zero since the conditioning happens on both $X^B$ and $X^E$, but $X^B$ is highly correlated with $X^A$, and hence, the residual uncertainty is almost zero.

Since the upper bound and the lower bound are equal, when Alice and Bob share highly correlated samples that the SKR reduces to $h(X^A|X^E)$; therefore, it is expected that for a high correlation, similar values for the upper and lower bounds should be achieved.

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
