# Peer review of "A Study on Secret Key Rate in Wideband Rice Channel"

_electronics, doi:10.3390/electronics11172772_

Round 1

Reviewer 1 Report

The current manuscript agrees on the compatibility issue of the existing state-of-the-art cryptographic solutions with the IoT limitations from the physical and energy perspectives. Studying the security aspect of the physical layer is an important topic, and the current paper is trying to address that. Unfortunately, I believe the current work would be great if the LWC (Lightweight Cryptographic finalist of the NIST lightweight for tiny devices competition) were mentioned in the current work as a novel solution for such IoT limitations. Moreover, the current position is a little bit far from the journal's scope as it is marked as an electronics journal. 

Author Response

The current manuscript agrees on the compatibility issue of the existing state-of-the-art cryptographic solutions with the IoT limitations from the physical and energy perspectives. Studying the security aspect of the physical layer is an important topic, and the current paper is trying to address that.

Thank you for the overall positive comment and for the time devoted to review our manuscript.

Unfortunately, I believe the current work would be great if the LWC (Lightweight Cryptographic finalist of the NIST lightweight for tiny devices competition) were mentioned in the current work as a novel solution for such IoT limitations.

Frankly speaking we did not know about the competition sponsored by NIST, but we agree it can be usefully mentioned in the paper. A dedicated sentence has been included in the Introduction.

Moreover, the current position is a little bit far from the journal's scope as it is marked as an electronics journal.

Thank you for this remark. Although the journal title mainly refers to electronic issues, the manuscript has been submitted to the special issue "Security and privacy for modern wireless communication Systems" under the topic Microwave and wireless communications. We therefore believe the paper is clearly and fully relevant to the subjects addressed by the special issue.

Reviewer 2 Report

I divided my comments using the sections of the manuscript. The number between parenthesis in the beginning of some comments refers to the row in the manuscript that comment corresponds to.

Overal comments

The manuscript clearly have its importance. However, it is not clear in the text the contribution. It is left for the reader to figure out. As it is not always possible for all the readers to do that, I would suggest writing one paragraph in the introduction clearly explaining the gap in the literature and how this work fills this gap. Although I was able to identify the contribution, I feel that I has to read the whole work to finally understand.

I also highly recommend improving the bibliographic review. There are many recent works dealing with SKR in IoT devices. The analysis made here is relevant in this field, so it is interesting to bring such papers as reference to support the importance of the work being presented here.

The methodology can also be improved, bearing in mind the reproducibility. There are parameters, such as the correlation matrix used, that should be indicated. I will give details about that and other things below.

I also recommend doing an English review throughout the paper, as it can be significantly improved.

Specific comments are as follows:

Abstract

(4) It relies on wireless channel reciprocity to let two legitimate devices exploit the random-like nature of the wireless channel to agree about a secret key

Physical Layer Security is not only about that. It also provides mechanisms to secure communications without relying on cryptography, such as artificial noise, jamming, precoding and beamforming, just to mention a few examples. This is clear when the authors write about that in the Introduction section, but in the abstract, that sentence might be misleading.

(9) The sensitivity of the key generation process can be expressed by the Secrecy Key Rate

Please include (SKR) after mentioning secret key rate for the first time in the abstract, since you use it after that.

In the abstract, the difference between this work and other works that also evaluated the SKR considering some channel parameters should be explicit, so the contribution of the work is clear from the start.

1. Introduction

(14) Previous studies on SKR (e.g. [12]) are mainly focused on the feasibility of the key generation process under the following conditions

The work in [12] is not about that. It is about estimating the Mutual Information required to compute the SKR. Hence, it is reasonable to consider, in [12], the conditions presented. There are, however, other works that consider scenarios with Line of Sight, correlation between samples measured in the same user (which degrades the key randomness) and also decorrelation between channel samples in different users due to non-simultaneous measurements. None of these works were presented or discussed here. They should be mentioned, and the difference between the present work and them should me explicit.

I would highly recommend improving the state of the art in the introduction. There are several works regarding SKR for IoT devices, very recent papers, dealing with other aspects such as quantization, improving key randomness. Including them in your bibliography review would improve the paper and help you highlight the importance of the work being presented here.

2. Physical Layer Key Generation Protocol

(95) Different quantization scheme can be used: uniform or non-uniform, single or multi level, differential-based or mean value-based [14].

The reference in [14] does not provide differential-based quantization algorithms. There are more recent papers that can be cited in this case. There are also papers that can be cited for the other types of quantization mentioned. This would make it easier for the reader to refer to these mechanisms and understand them.

Figure 1 is not referenced anywhere in the text.

(110) Previous works focused only on the mutual information between Alice and Bob [12], simply neglecting the presence of Eve

This is not true. There are works that consider the presence of Eve. I suggest a more throughout review of the literature.

3. Materials and Methods

(132) lot of channel realizations are generated for the same input values

Instead of “lot of”, a number should be specified, even though it is explicit in Table 1 (which was not mentioned until this part of the text). Avoid expressions such as “a lot of” in these cases. They make the text vague.

(141) The wireless channel is described by means of the "TDL-D"

The acronym TDL-D should be specified the first time it is used.

(166) The acronym CTF should be defined the first time it is used.

(172) this method is might still fairly work to generate correlated Rice channels, if it is applied to the square of the channels

As the method is applied to the PSD coefficients, does the PSD coefficients follow the Nakagami-m distribution (used to approximate the Rice distribution)? Until now, we only know that the envelope of the channel impulse response follows the Rice distribution, but no consideration is made in the text about the distribution of the coefficients of the PSD.

Further, what is the desired correlation matrix used? Was it estimated? How was it obtained?

4. Gaussian Case and Validation

I worry, although important, that this section only shows the validity of the method for the Gaussian case. The authors do not show that X_i is Gaussian. Perhaps, looking at equation (6), the authors can use the Central Limit Theorem to show that the distribution of X_i is Gaussian. Since it is an integral of the PSD, (sum of random variables with the same distribution will result in a random variable with Gaussian distribution). If it can be shown that X_i follows a Gaussian distribution, then this section will be sufficient to show that the method is valid. Nonethele, this argument should be included in the text.

I would also like to suggest a modification in Figure 4. It would be interesting to use different line styles for the theoretical and estimated curves. For example, theoretical curves solid, as it is now, but the estimated using doted or dashed lines. This way, we would be able to see the superposition of the estimated curve with the theoretical one.

(231) the estimation has been done both for 1 filters (narrow-band case, Figure 5) and for 4 filters (wide-band case, Figure 6).

What are the frequencies f_i used for each filter, and other parameters, such as \Delta t?

5. Results and Discussion

The acronym DF should be defined the first time the term “delay spread” appears in the text (row 226). The term delay spread was used several times in the text, but suddenly it was defined as DS on row 264. Define it the first time it appears, and only use DS after that.

Author Response

Thanks for the useful comments, we replied to all the comments in the attached file

Round 2

Reviewer 2 Report

The authors have addressed all my considerations.

I would recommend, just for the sake of completeness, to include the parameters of the distributions used for fitting the PSD. It can be included directly in Table 2.

Author Response

Thanks for the suggestion, the parameters of the distributions have been added in table 2 as suggested. The notation is the standard scipy.stats notation with the shape parameter, the loc and scale as indicated in the documentation of the distriubtion.